# A Comprehensive Review on E-Waste Management Strategies and Prediction Methods: A Saudi Arabia Perspective

**Hatim Madkhali [1,2], Salahaldeen Duraib [2], Linh Nguyen [3], Mukesh Prasad [1], Manu Sharma [1,4] and Sudhanshu Joshi [1,5,\***

1   School of Computer Science, FEIT, Australian Artificial Intelligence Institute,
    University of Technology Sydney, Sydney, NSW 2007, Australia
2   College of Computer Science and Information Technology, Jazan University, Jazan 82817, Saudi Arabia;
    sduraibi@jazanu.edu.sa
3   Institute of Innovation, Science and Sustainability, Federation University, Churchill, VIC 3842, Australia
4   Department of Management Studies, Graphic Era Deemed to Be University, Dehradun 248002, India
5   Operations and Supply Chain Management Research Laboratory, School of Management, Doon University,
    Dehradun 248001, India
*   Correspondence: sudhanshu.joshi@uts.edu.au

**Abstract:** Electronic waste generation is increasing dramatically throughout the world. Consequently, this increase in E-waste harms the environment, health, and other aspects of human life. Moreover, hazardous substances and the informal disposal of E-waste severely threaten human health and the environment. Saudi Arabia is the largest Arab country in terms of electronic waste generation and is the Arab country that generates the most E-waste. Over the past few decades, several initiatives and policy implementations have been undertaken in the country. However, the management of E-waste is still a source of distress and an unresolved issue. Sustainable development requires much more effort, primarily efficient E-waste management, which can only be achieved by establishing a formal collection system, early forecasting, and accurate estimations. The purpose of this study is to provide an overview of the fundamental and emerging trends in E-waste production worldwide and in Saudi Arabia. This paper aims to summarize the hazardous elements present in E-waste, their dangerous effects, and the economic potential for recycling E-waste as a means of developing sustainable communities. This study explores the practices regarding efficient E-waste management and E-waste estimation and prediction globally, specifically in Saudi Arabia and other Arab countries. This study suggests that more than the use of a single management probe will be needed to achieve significant results. Instead, a complementary approach can be used to achieve the most effective results. Additionally, people should be aware of the importance of handling and recycling E-waste. This study emphasizes the importance of formal collection and documentation to ensure effective monitoring and sustainable development in any developed country.

**Keywords:** E-waste; E-waste generation; E-waste management; environmental challenges; health impacts; E-waste estimation; E-waste prediction; sustainable management; public awareness

## 1. Introduction

In recent years, urbanization has led to a tremendous increase in the use of electricity and electronic appliances. The widespread use of electric devices has raised concerns about generating electronic waste and its leftovers [1,2]. Electronic products and their composition, including hazardous materials, have become significant concerns due to their adverse effects on the environment and human health. The U.N. E-waste monitor classifies E-waste as a range of leftover products that contain any circuitry components and require electrical power to perform their function. Six basic categories of electronic waste fall under these general categories. Figure 1 depicts the composition of E-waste.

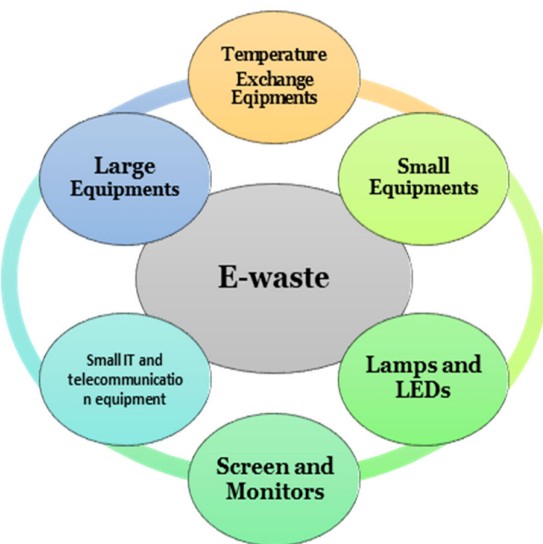

**Figure 1.** Composition of E-waste.

Due to the heavy metal ion composition contained in E-waste such as Cd, Pb, Br, Hg, Cr, and other flame retardants, it is very dangerous to humans and the environment in many circumstances when in direct contact with this kind of waste [3]. The improper use of landfilling and water treatment is causing ion leaching and the contamination of natural resources (air, water, and soil) [4]. These hazardous materials can destroy humans' cardiovascular, respiratory, digestive, and even neurological systems on a direct interaction basis [5,6]. Furthermore, using highly concentrated acids and cyanide to leach and recover costly metals also poses serious health risks [7,8]. While there is a large amount of consumption and demand for E-waste, it cannot be monitored appropriately, as suggested by a recent study which indicates that approximately 7–20% of electronic residuals are exported as secondhand items (e.g., metal scraps) or imported [9].Thus, secure, cost-effective, environmentally sound, and sustainable E-waste management and handling must be employed, including proper recycling, metal recovery, transportation, and the disposal of E-waste. This will eventually help in the mitigation of environmental hazards as well as balance out the natural resource distribution [10,11]. A proper E-waste handling framework and ecosystem approach is required for collection, separation, and consumption or recovery at national and global levels. In terms of the national level, this framework must incorporate local-level management to curb the malpractices in E-waste disposal. Due to the presence of precious metals, E-waste should be treated physic-chemically to recover any metals which generate toxic plastics and fumes, as well as brominated flame retardants (B.F.R.s) and liquid crystals (E-waste Monitor, 2021). As in the case of recycling, the plastic of "end-of-life printers" can be utilized as reusable carbon, whereas the waste of tannic powder can be further processed to get iron, thereby reducing the waste production and disposal costs as well as facilitating a closed loop in a circular economy within the sector [12]. Nowadays, the rate of urbanization, industrialization, and the development of modern technologies are all affecting the country's economy, which cannot be sustained without the generation of electronic equipment [13–15]. Here, the ecosystem approach is advocated as a sustainable and environmentally-friendly way to plan for the disposal of E-waste and the reclamation of noble metals [16]. For instance, consuming pure concentrated acids, cyanide, and nitrox during informal metal recovery causes severe human health and environmental hazards [17]. Therefore, cheap, secure, green recycling and recovery approaches are critical when handling E-waste today. Due to the short life span of e-products, the retrieval of noble metals can benefit the economy [18]. Only a few recycling plants have deployed this approach in the countries where the waste is generally shipped. As a result of poor or disrupted engineered infrastructure, as well as the quite costly management processes, most nations are currently managing their E-waste informally

without ropes and safe metal recovery techniques [19,20]. The inadequacy of available resources has worsened the situation [21]. Despite ongoing research and discussions on the production and handling of E-waste since the late 20th century, the problems associated with it persist. [22,23]. The literature review indicates the tremendous and rapid E-waste growth, which has caused stakeholders problems regarding its management and disposal [24]. The problems have also increased due to the accelerated rate of the generation of e-products. This research aims to highlight the global aspect of E-waste generation and management along with its regional and national impacts. The E-waste management approaches and constraints when managing E-waste, along with forecasting future E-waste production, have been highlighted and used to recommend suitable strategies and best practices for handling and regulating E-waste. This is intended to result in a significant reduction of negative environmental and well-being impacts [25]. This research is aligned as follows: it begins with the introduction, followed by the global E-waste scenario and Saudi Arabia's perspective. The problem statement, the impact and significance of this study, and the various assessment methods follow. The methodology and exploration of E-waste forecasting and its estimation follow this. Finally, the research ends with recommendations for future work and concluding remarks on better managing E-waste production and recycling. Amongst others, the life cycle assessment (L.C.A.) and material flow analysis (M.F.A.) are the most adopted and practiced assessment tools used to quantify the several dimensions of the E-waste organizational system along with metal restoration in a circular economy [26,27].

## 2. Research Methodology

It was necessary to conduct a detailed literature review to characterize electronic waste, its handling, recycling and prediction, and the emerging issues involved in the field. The article insight and exploration were conducted using search engine tools such as Scopus, Web of Science, PubMed, Elsevier, Springer, Emerald, Cell Press, BioBacta Google Scholar, and Wiley. Supplementary article research was also carried out by hand, searching the citing articles. All cited articles were input into the citation manager software to screen them according to the PRISMA instructions in the light of Kitchenham et al. [28] and Liberati et al. [29]. Several keywords and strategies were chosen to avoid biased research. The articles were screened using the research method defined during the literature survey. After reviewing the articles' titles and abstracts, several articles were selected for a more extensive study. After a detailed study, some of them were again omitted and the rest were adopted for inclusion in this systematic review. The detailed analysis of the conducted research and literature survey led to the exclusion of 55 repetitive and irrelevant articles that did not focus primarily on waste. Through this methodology, the aim is to understand the country's E-waste production and correlate it with regional and global production, and how efficient E-waste management can help maintain a sustainable environment.

Furthermore, the adopted model for estimation and forecasting was explained to help. This study focused on sustainable management's economic, environmental, and social aspects. This is why we have discussed the supply chain network, consumer behavior as well as the impact of social awareness. It was determined during this research that more documentation needs to be made regarding E-waste production, collection, and processing. Overall, this article provides an overview of E-waste in Saudi Arabia, its corresponding challenges, and the opportunities available.

## 3. Global E-Waste Generation

Global E-waste monitoring (Figure 2) has determined that in 2021, 53.6 million metric tons of E-waste was generated globally, with a mean increasing rate of 2.5 Mt per year. In 2014, the E-waste generated was recorded as 44.4 Mt, which shows that there has been more than a 20% increase since 2014. This indicates threatening circumstances [30]. Among all of the studied countries, the USA is the highest E-waste generator, producing 13.1 Mt of E-waste annually, while its per capita involvement is 13.3 kg. However, only

~1.2 Mt of the whole of the E-waste is properly managed, i.e., formally gathered, monitored/acknowledged, and reprocessed as per the U.S. Environmental Protection Agency (USEPA) instructions. The USA has indicated that, on average, more than 20 items of electronic equipment are consumed by an American household [13]. Europe has shown itself to have the highest sustainable and formal E-waste management, i.e., 42.3% (5.1 Mt) of the produced E-waste or 12 Mt. Asia has produced the highest amount of E-waste at 24 Mt and recycled only 2.9 Mt. Europe tops the list of E-waste generation per capita in the world (16.2 kg). China is among the highest E-waste producers at 10.1 Mt, while India is the highest producer (3.2 Mt) after the USA and China. However, its per capita contribution is relatively low (2.4 kg) compared to the global one, i.e., 7.3 kg. Over the past nine years, E-waste production has surged by 58% and accelerated [31].

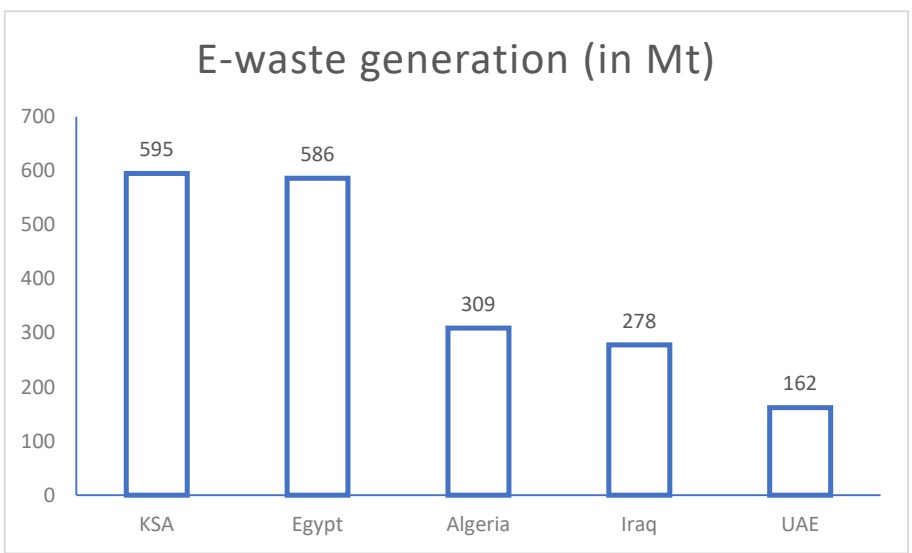

**Figure 2.** Global E-waste generation (in Mt).

The data collection regarding E-waste management shows that only 44.3 Mt of E-waste is handled formally, less than 40% of the global production. Most of the E-waste of large income-generating countries is shipped to progressing or lower-income countries. Most household electronics end up in the trash, which is also the primary reason for excluding E-waste from formal collection and documentation. Records show that approximately 0.6 Mt of E-waste finishes in trash boxes. In the coming decade, waste generation is projected to accelerate drastically to 121% from 2020 to 2030 [32]. The high projection and non-linearity are due to the correlation between G.D.P. and E-waste generation, which shows that the saturation of E-waste generation is related to moving towards being a higher economic wealth country from a lesser one [33,34].

Most European countries had a relatively larger waste generation per capita (20–25 kg/capita) than others in 2019. In the case of Asian countries, India had the lowest E-waste production per capita (~2 kg/capita), perhaps because of its inhabitant density with respect to E-waste production [35–37].

### 3.1. E-Waste Generation in Arab Countries

Over the past ten years, E-waste production has increased by more than 60% in Arab countries, from 1.8 Mt to 2.8 Mt, with a 0.1% management rate. EEE-POM has surged up to 30%, totaling 3.2 megatons (Mt/8.8 kg/Inh) in 2010 to (4.1 Mt/9.5 kg/Inh) in 2019. The largest E-waste generation per country is (595 kt/13.2 kg/Inh) of E-waste, increasing up to (0.6 kt/0.7 kg/Inh), showing significant diversity in the region. The E.E.E. production is relatively low, whereas E-waste generation and EEE-POM are interlinked positively with G.D.P. Increasing the G.D.P. for P.P.P. resulted in an increase in the EE-POM for Arab countries. The EEE-POM per inhabitant is the largest in the case of Qatar (24.9 kg/Inh),

while it is lowest at 0.8 kg/Inh in Comoros. In terms of absolute value, Egypt is the second largest E-waste generator. It has the highest EEE-POM ranking at 1.1 Mt, followed by Saudi Arabia (758 kt), Iraq (459 kt), and Algeria (458 kt) [36,37]. Figure 3 depicts the E-waste Production in Arab Region.

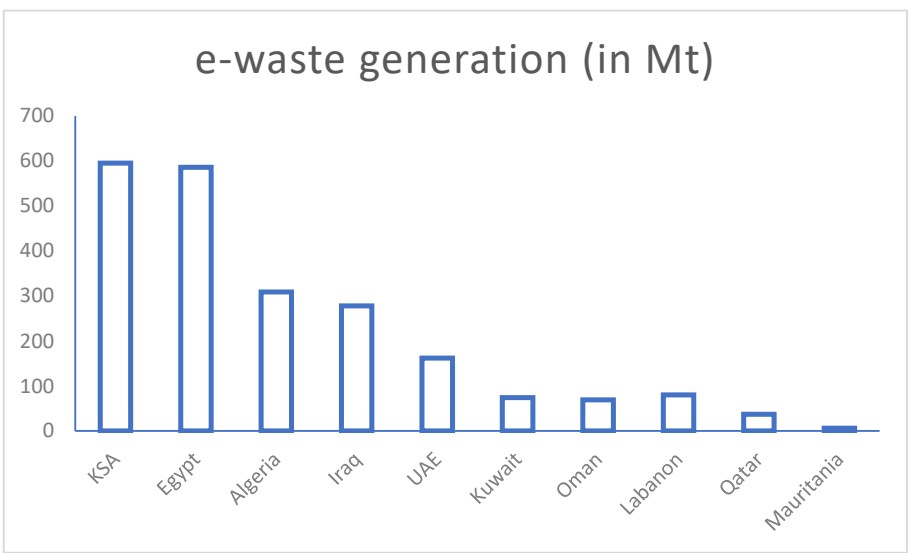

**Figure 3.** E-waste Production in Arab Region.

### 3.2. E-Waste Scenario in Saudi Arabia

According to the United Nations' E-waste monitoring, the Kingdom of Saudi Arabia is the highest E-waste producer among the Arab countries with 595 kt (or 16.3 kg/Inh), i.e., 21% of the region. The EEE-POM value of the Kingdom is 758 kt, the second largest after Egypt [35,36]. The relationship between EEE-POM and G.D.P. shows that E-waste production is increasing tremendously with improved living standards, i.e., access to technology, the I.T. revolution, and urbanization in Saudi Arabia. Currently, 22% of the produced hazardous waste is treated, while the rest is informally treated or leftover [36]. In the past four years, almost 40% of the waste production has increased in the country [37]. Despite the huge amount of E-waste, laws and policies regarding E-waste management were not developed until 2017. In 2022, the introduction of the National Transformation Program (N.T.P.) 2020 was intended to lessen the amount of E-waste by 40%. Through the waste to energy and sustainable environment initiative of Saudi Arabia's Vision 2030, the aim is 100% E-wasted recycling [38]. Currently, there is poor awareness regarding E-waste [39,40] and technological hurdles are making sustainable management impossible [41,42]. The current waste management practices involve collection through bins individually and disposal in the landfill, which is 10–15% of all waste [43,44]. This is due to the informal waste management system prevalence in the country. However, some waste management companies rely on manual sorting without using a high-tech recycling facility specifically for E-waste. Furthermore, data regarding the regulation and composition of E-waste has yet to be documented. On the other hand, the USEPS has predicted the largest E-waste production rate among all Arab countries will be in Saudi Arabia. According to this, if the annual E-waste increases from 5% to 10% in terms of E-waste due to the increase in inhabitants, improved livelihood, and industrial advancements, then in 2040, E-waste will range from 1345 kt to 4507 kt [45–48].An ecosystem approach is needed for efficient and sustainable E-waste management that covers the environment, human behavior, social attributes, and the economy [49]. This is shown in a case study in Romania which helped identify hot spots, high-performance areas, and the methodology performance. WEEEs technical components and sustainability assessment are organized based on the social, economic, and environmental aspects. In case some indicators are qualitative and quantitative; the indicators are developed to cover the environment, policies, resource management,

financial soundness, and user inclusivity using national-level E-waste statistics, economic numbers, and consumer behaviors [50]. Using models for correct E-waste anticipation and to increase preparedness can also help in the short term as they involve the early prediction of E-waste generation in the region. For example, grey model E-waste forecasting allows for a novel and accurate prediction [51–53]. Comparative analysis of international E-waste management models leads to the conclusion that the current situation of E-waste production in Saudi Arabia is posing a serious threat that demands a concrete and strong E-waste managing and regulating framework covering the local level of waste collection through to the cross-boundary exchange. To increase the impact of said model, social awareness and the enhancement of the seriousness of the problem among the people can help to achieve better results [53]. The Extended Producer Responsibility (E.P.R.) and Value-Belief-Norm Theory (VBNT) are the most adopted and practiced frameworks for behavior analysis in E-waste management [54–57] that will help to achieve the efficient conversion of waste prior to landfilling, thereby enhancing the management of E-waste disposal and environmental sustainability [58].

## 4. E-Waste Supply Chain Network (S.C.N.)

For economic circulation as well as sustainable E-waste management, it is mandatory to consider the supply chain. Supply chain networks comprise two major components; one informal and the other formal. The informal one is considered to be on the micro-scale. It is highly labor-consuming, mostly unregulated, and involves poor technological utilization or the provision of resources and services. The formal sector dominates due to the monitoring and regulatory opportunities. This is the sector that contains the most information in Saudi Arabia, as most of the E-waste is produced and regulated in the informal sector [59]. Improper management techniques such as acid leaching, dismantling, and improper landfilling make sustainable management difficult, while their existence and widespread nature make the adoption of formal recycling [60]. The multifaceted nature of E-waste causes the handling and segregation, along with the dismantling, to be more difficult as well [61]. E.P.R. is considered to be the most feasible approach, but it has a weak implementation framework [62]. E-waste management can be achieved by considering the following factors: the analytical hierarchy process (A.H.P.) along with quality function deployment (Q.F.D.) for decision-making, evaluations, and system management [63,64]. Using both structures will help to regulate the predefined critical factors, enhancing the efficacy of the S.C.N. The largest e-generator handling systems are being managed by this model, which shows the significance of this process. Moreover, a study has suggested that an increase of $1000 in G.D.P. will increase the amount of E-waste generation by 0.5 kg. This clearly demands sustainable and dynamic management [65].

## 5. E-Waste Management Policies and Strategies

Several strategies have taken place in this field. These strategies include E.P.R., L.C.A., M.F.A., M.C.A. [66], and VBNT. Some countries have used analytical management tools [67] and GIS-based [68] systems for E-waste management. However, only some tools can be considered to be the prime method. Collectively, these tools can complement each other's impacts. The success of these models depends upon the environmental footprint, secure recycling, and the recovery of precious metals through formal collection and segregation protocols.

The life cycle assessment (L.C.A.) has gained global acceptance, with a particular emphasis on Europe, Asia, the North-eastern and African regions, as well as Switzerland. In these areas, extensive research has been conducted on the eco-friendly and economic benefits of recycling systems [69] Asian countries have adopted the L.C.A. mostly for the estimation of electronic waste management and its impact. This has shown significant economic and environmental results [70]. In comparison with conventional approaches— i.e., incineration, landfilling and the disposal approach—L.C.A. is the most significant E-waste handling strategy [71,72]. In India, when using the L.C.A. model, E-waste prediction

and its estimation is carried out using software, tools, and E-waste datasets which are then compared with real time datasets. These studies have shown the significance and reliability of the L.C.A. predictions [73].

On the other hand, material flow analysis (M.F.A.), a decision support system, is another significant tool used to spatially analyze the flow and timeliness of E-waste during recycling, E-waste collection, disposal, and the stocking route. In this way, M.F.A. provides information on the interface linkages from the initial stage to the destination [74] This includes considering the flow of E-waste and its assessment in terms of environmental, economic, and social value. India, China, Indonesia [75], Australia [76], Japan [77], and Nigeria have adopted the M.F.A. tool to map the flow of E-waste, as well as its collection, recycling, and cross-boundary trade [78]. E-waste generation and estimation in the USA have also been examined using the M.F.A. model using previous and future electronic gadget sales details and their average life span. This has provided the baseline for the end of the life span information of electronic components, and has linked the EoL with E-waste generation and management. Therefore, it is suggested that by decreasing the EoL of electronic computers, E-waste generation can be decreased [79]. M.C.A. is suggested to be the best tool to incorporate social enablers in E-waste handling [80]. This decision support system has been adopted by the USA, Spain, and Cyprus to optimize and locate recycling plants and E-waste handling. This tool comprises two phases which enable the evaluation of any constraints and issues followed by efficient data processing. In this way, a balance between the environmental and economic footprints can be achieved. Among these tools, E.P.R. is a national approach that provides information on the environmental aspects and directs the responsibility of the manufacturers to take back products after they have reached the end of the life phase [81].

Sustainable E-waste management can only be accomplished following an accurate analysis of the existing E-waste, its generation rate and future prediction. For this purpose, quick predictor models such as Bass, Gompertz, Logistic, ARMA, and the Exponential Smoothing model have been utilized to determine future E-waste generation; the content of precious metal ions in computers in Japan, Australia, New Zealand, the USA, and South America were examined. These models have offered reduced error indices, reliable in-sample estimation, baseline mapping, and a framework for E-waste estimation [82]. Most countries follow the Basel Convention and Stockholm Convention 2001. Along with these, digital technologies are also emerging as promising tools for E-waste management. The 4.0 industrial revolution has offered several socio-economic and environmental solutions to reduce pollution, curb waste generation, and improve working conditions by adopting smart vehicles, robots, A.I. tools, real-time monitoring, and sensors [83]. With these emerging robotics, smart tracking systems, sensors, RFID tools, smartphone apps, real-time monitoring, and autonomous systems, sustainable and eco-friendly E-waste management is achieving remarkable outcomes [84]. Although robotics bins, intelligent routing, autonomous cars, and real-time monitoring are practiced interventions, they require further ecosystem development to achieve optimal results [85]. In Turkey, triple bottom line (TBL) technology has been adopted to digitalize E-waste management and its prediction. In this process, E-waste generation is predicted for the coming four years, and then a novel sustainable collection, characterizing, and segregation center model is proposed. The model is based on digital technologies using the TBL concept according to the projected increase of E-waste. This concept can further be utilized by municipal companies and other authorities showing the center model's implacability [86].

The E-waste management in Gulf countries relies on existing laws and compliance with the Basel Convention, as no specific regulations exist. However, the United Arab Emirates has recently implemented EPR-related laws, and Jordan, Kuwait, and Algeria are also in the process of enacting similar legislation. Other countries in the region have yet to introduce any laws or regulations pertaining to E-waste management [87].

In addition, article 46 of Egypt's Telecommunication Regulation Act (No. 10/2003) prevents importing utilized telecommunication technology probes for interchanging [88].

For E-waste management, the adopted activities are often manual, informal, and labor-intensive. Under the Vision 2030 policy and sustainable environment initiative, composting and waste-to-energy opportunities are gaining increased attention in the country. Composting is considered due to the high organic waste generation (around 40%). All E-waste management activities are coordinated and financed by the government. The most practiced approaches in the country are currently landfilling and incineration [89].

## 6. E-Waste Recycling and Metal Recovery

Currently, the most appropriate physical processes used as pre-treatment techniques according to the technological aspect are:

- Pyrometallurgy;
- Hydrometallurgy;
- Biohydrometallurgy;
- Pyrolysis.

Non-chemical approaches are now becoming the most feasible approaches prior to the recycling or treatment of E-waste [90]. The pyrometallurgical methods involve the use of magnetic separation, eddy currents, air currents, and vacuum metallurgical separation [91]. Hydrometallurgy uses chemicals, whereas bio-hydrometallurgical uses "green technology," i.e., microorganisms for metal abstraction [92,93]. Using microorganisms to regain noble metals is a relatively economic and environmental approach regarding several other resources [94]. The thermal cracking (pyrolysis) or thermal conversion of E-waste is quite a beneficial and emerging technology. However, it cannot be implemented at a larger scale due to restrictions regarding thermogenetic and initiation energy and yield [93,94]. These sophisticated adopted technologies are indispensable for better recycling and environmental sustainability. They are on hand to help restore natural resources while also reducing the hazardous and economic burden [94]. The recovery scheme involves the use of mechanical separators to sort, crush, and separate electronic parts, followed by pyrometallurgical treatment that deals with nonferrous metals and separation of metallic and noble elements and the separation of metallic and noble elements. For metallic item recovery, electrometallurgical treatments are carried out, followed by a treatment with liquid in which the acidic wastewater is neutralized and treated before discharge [95].

## 7. E-Waste Estimation and Prediction

Presently, many strategies have been established for the efficient and eco-friendly handling of E-waste management worldwide, specifically in developed countries. Table 1 summarizes the adopted technologies used to propose specific interventions. A detailed discussion is then carried out below the table.

Several studies have already discussed E-waste generation and disposal. One of these studies performed a case study to estimate the E-waste generated by abandoned cathode ray tube C.R.T. devices in Germany. The data was collected via a survey, engaging both the recycling centers in the country and residents to determine the product lifetime distribution. The number of household devices was collected from published statistical data. A use-phase analysis forecasting method was used to predict future waste based on the collected data [96]. Another researcher performed a survey in Vietnam, collecting the raw data and then using it to assess the amount of E-waste generated by five different kinds of electronic equipment. The number of disposed appliances was determined using a population balance model [97]. A survey approach was used to predict the amount of E-waste from eight different electronic devices in South Korea. The data was collected through a questionnaire, and Weibull distribution was used to analyze the life span distribution of the eight appliances [98,99] used Holts' double exponential smoothing and dynamic life-span technique to estimate the E-waste from 16 electronic devices in the Australian environment from 2010 to 2030. The authors stated that the Weibull distribution prediction model is the best method out of a group of ten prediction models to use to predict the future amount of E-waste. The data was collected from an open source, the passport database

"Euromonitor." In practice, such a technique has proven to be beneficial when analyzing time series data connected to EE-waste creation. According to the article, future work could explore another database [100]. One of the efficient ways to engage in recycling and E-waste treatment involves precise estimations using a grey model in which fractional calculus and integral-differentiation operators are used as non-local probes, linking with the E-waste management both in the present and historically which, again, can be broadly applied. In a study looking into the generation process of an E-waste model, it is suggested to enhance the integral model through a differential equation; i.e., a fractional grey model which not only predicts the E-waste, but also offers data on the waste volume and number of metals. This type of model is a non-singularity, involves easy and simple computation, and presents a feasible solution with accuracy and precision [101]. In another study, the grey modeling technique was adopted for E-waste prediction in Washington, USA. This provided reliable E-waste forecasting and estimation. Here, the Nash nonlinear grey Bernoulli model was integrated along with fractional locators using PSO. The specifics were created with the intent of facilitating municipal department usage and reverse logistic operations, thereby dispersing the data employed by formal E-waste management in the years ahead.

**Table 1.** Summary of the E-waste prediction and estimation methods and strategies.

| Author | Year | Application Purpose | Tools | Proposed Intervention |
|---|---|---|---|---|
| Hischier et al. [69] | 2005 | E-waste handling | L.C.A. | Developing economically and socially favorable E-waste strategy developing tool |
| Kazancoglu et al. [85] | 2020 | Digitalized E-waste tracking and handling | M.F.A. | Decision Support system for better E-waste handling |
| Yoshida et al. [77] | 2016 | Mapping E-waste flow | M.F.A. | Better E-waste collection and cross-boundary trade |
| Kiddee et al. [66] | 2013 | E-waste estimation | Decision support system | Developed the linkages during the interfaces from the initial to the final stage in E-waste management. |
| Andarani and Goto [75] | 2014 | Mapping E-waste flow | M.F.A. | Better E-waste collection and cross-boundary trade |
| Kiddee et al. [66] | 2013 | E-waste collection | E.P.R. | Responsible and formal E-waste collection |
| Roychoudhuri et al. [73] | 2019 | E-waste prediction and estimation | L.C.A. | Model predictions and comparison with real-time data |
| Forti et al. [83] | 2020 | E-waste estimation | Smart technologies | Real time monitoring, estimation and forecasting |
| Islam and Huda [76] | 2020 | Monitoring E-waste flow | M.F.A. | Better E-waste prediction |
| Duman et al. [92] | 2020 | E-waste prediction in Washinton U.S.A. | Grey Model | Provided reliable forecasting and E-waste estimation based on an open dataset in the USA |
| Andeobu et al. [25] | 2021 | Forecast E-waste generation in USA, and UK. | V.M.D., ESM, G.M. | Predicted the future fluctuation trends of E-waste production which helps in the proposal of timely interventions and decision-making as part of the sustainable circular economic goals. |

The significance of this study is that the reliable and accurate information uses minimal datasets and also provides a comparative analysis within the E-waste data itself [102].

In China, the Sales Obsolescence Model technique has been adopted for E-waste estimation and prediction. During this estimation process, the E-waste generation datasets

revealed that 8 million tons by weight of E-waste are produced domestically, with 1% of the metal retrieved during the recycling process. This study, along with the E-waste estimation, suggests the development of an urban mine for critical metals during the E-waste recycling processes [103]. A recent hybrid decomposition-ensemble model integration approach—which integrates variational mode decomposition (V.M.D.), an exponential smoothing model (ESM), and grey modeling (G.M.)—has been used to forecast E-waste generation in the USA and the U.K. The most significant prediction result for E-waste data also predicts the future fluctuation trends of E-waste production, helping to engage in timely intervention and decision-making to better meet the sustainable circular economic goals [104]. Another study used regression analysis to estimate the volume of E-waste created across the Indian industrial sectors in order to establish a regulatory framework [105]. Moreover, the distribution delay model was employed to forecast mobile phone End-of-Life in the Czech Republic [106]. Aside from this, another study attempted to anticipate the amount of E-waste generation in the USA by utilizing the material flow analysis (M.F.A.) approach. This method was used to anticipate the future E-waste of thirteen E.E.E.s. Historical and future sales records are required to apply this model and the life span assumption [107].

Furthermore, the M.F.A. approach was used with a logistic model to anticipate the number of outmoded P.C.s [107]. Additionally, grey modeling was applied to estimate E-waste generation in Botswana [108]. Another study used a different technique by employing discrete grey modeling to quantify the quantity of E-waste created by electronic products like televisions, mobile phones, and PCs in the Indian environment. In this study, the forecast was done using a discrete grey model that incorporates the Fourier transform and exponential smoothing techniques [109]. Grey modeling has been used also to estimate the amount of E-waste collected in Turkey [110]. This research demonstrates that grey modeling may yield valuable insights even with small, limited datasets. The reliable accuracy of the models varies from one scenario to another and based on the characteristics of the databases. For instance, grey forecasting models are renowned for providing accurate predictions whenever used with small-sized datasets. Among several prediction models, the univariate grey forecasting model GM (1.1) and its improved variants have become popular due to their improved accuracy. In addition, higher precision grey models are often attained by adjusting the model's parameters [111].

## 8. E-Waste Inventory

E-waste management also demands the tracking of the in-outflow of electronic products for developing strategies, infrastructure, and a sustainable supply chain network, including the economic, social, and environmental aspects. Hence there is a need for a dynamic and extensive database. This can be achieved through the development of a detailed E-waste inventory. In this regard, a study has been carried out in India on the development of an E-waste prediction and inventory model for households in Pune city. The model provided segregation, transportation, tracking, and data storage for use regarding the cooperative functionalities available and reducing hazardous waste. Thus, it offered sustainability, reliable solutions, and new economic propositions for the stakeholders and end-user engagement [112]. A study carried out in Pakistan declared the development of an inventory to be imperative for developing countries as it is in these areas that E-waste generation is in a state of fluctuation and increasing tremendously. They also showed the current expansion of E-waste production, which clearly demands proper E-waste handling and disposal [113]. In India, two suggested methods are E-waste inventory determination through a questionnaire-based survey, and a Cloud-based model. The consumer buying capacity and life span detail the utilization of E-waste inventory to provide the in-out flow of e-products which is then linked with E-waste generation. This facilitates smart E-waste inventory. There is also the data available on e-product buying and selling, which can be further utilized in decision-making and strategy development [114].

## 9. Recommendations and Future Work

E-waste management has become unattainable because of its collection, segregation, and recycling, depending on the type of waste. However, it is necessary for sustainable development and the environment [115]. With the technological advancements and research developments that have taken place, specific methods for recycling and recovering have been identified, and the fusion of these methodologies with various forecasting, estimating and analytical methods will help to achieve efficacy and promising results. The significant point in this domain is the availability of superfluous comprehensive information within a detailed reviewed analysis which can serve as a pillar to implement newly developed systems [116]. In this case, systematic research and a holistic approach can be helpful shortly [117], along with the tracking and documentation of E-waste collection, segregation, and recycling [118]. To enhance the anticipation and adaptive capacity of people toward E-waste collection, the government, private television, print media, and social networks can be utilized [119]. During E-waste tracking, real-time monitoring, commands, and data reserving to process the variables and ensure the implementation of the safety protocols and the safe storage of waste must be ensured for eco-friendly and sustainable management. Accepted physical and chemical treatments must be carried out during waste management [120]. Isolated regions must be designed and advertised to make the public aware of the severity of the problem. In addition, the following perspectives have been drawn on from this study [121–123]. The optimized supply chain network, along with extended consumer responsibility (E.C.R.), must be developed as part of this waste management regulatory environment [124]. During management, the recycling systems' economic soundness, security, and feasibility must be evaluated to make them more environmentally friendly.

### 9.1. Specific Solutions

Ikhlayel [125] proposes an integrated method for improving E-waste management in developing nations by addressing region-specific concerns. To verify the model and its assumptions, the technical features of this methodology were contrasted with previous approaches. Nowakowski, Szwarc, and Boryczka [126] presented a precise answer to the issue of E-waste buildup. The solution included a combination of artificial intelligence-assisted collection and a novel vehicle body, a harmony search algorithm for route optimization in on-demand E-waste collection, convenient loading logistics, efficient waste collection, loading, vehicle routing, and packing, and improved health benefits to the population due to waste removal from their premises.

### 9.2. Effectiveness of Various Models in Future

In a case study conducted in Jordan, a comparison of five E-waste management strategies for six E-waste items, 506 showed that integrated trash management is the best solution. The integrated waste management system 507 includes recycling materials including non-PMs and PMs, incinerating plastic and the harmful components of P.C. Bs using energy recovered from incineration, and using sanitary landfills for residues. Under this circumstance, the 510 treatments of cellphones had the optimum environmental performance. Energy-recovered incineration of a part of hazardous waste is a 511 alternative worthy of careful investigation. Reducing the amount of E-waste and recycling were proposed as the optimal solution for the Chinese scenario [127].

### 9.3. Work to Develop Clear Understanding by Decision Makers

Research has demonstrated that creating efficient and effective E-waste management systems is feasible. Numerous successful models with solutions have already been addressed. Decision makers may help fund studies comparing these models in their nation. Decision makers may choose the best approach for E-waste management in their nation based on the findings of comparison studies (including their relative economic feasibility).

## 10. Conclusions

Anthropological activities generate an increasing amount of E-waste, and poor management strategies without formal collection, segregation, and a recycling system make E-waste management more difficult. This will eventually affect environmental sustainability and cause harmful impacts on health and the natural ecosystem. Current practices in Saudi Arabia lack efficiency and segmentation, requiring an adequately regulated network with early estimation and prediction and concrete actions to mitigate the harmful impacts. Therefore, developing a regulatory framework with monitoring and estimation is highly recommended, along with raising social awareness. This integration will help to control and manage E-waste in an integrated and secure system which will eventually curb the negative impacts and enhance environmental sustainability.

**Author Contributions:** Conceptualization, H.M. and M.S.; methodology, S.J. and M.S.; Software, H.M., S.J. and S.D.; Validation, S.J., H.M. and L.N.; Formal analysis, S.J., S.D., M.P. and M.S.; Investigation, M.S.; Resources, H.M., S.J. and L.N.; Data curation, M.S.; Writing—original draft, S.J. and H.M.; Writing—review & editing, M.S. and S.J.; Supervision, M.P.; Project administration, S.J., H.M. and M.S. All authors have read and agreed to the published version of the manuscript.

**Funding:** This research received no external funding.

**Institutional Review Board Statement:** Not applicable.

**Informed Consent Statement:** Not applicable.

**Data Availability Statement:** Not applicable.

**Conflicts of Interest:** The authors declare no conflict of interest.

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
