# Peer review of "A Comprehensive Review on E-Waste Management Strategies and Prediction Methods: A Saudi Arabia Perspective"

_knowledge, doi:10.3390/knowledge3020012_

Round 1

Reviewer 1 Report

E-Wastes  is of great concern nowadays. E-waste management strategies are very important for  the recycling of these important resources.  In this review, a complementary approach are proposed to achieve the most effective results. The review is well organized generally and can provide the readers meaningful information. It can be accepted after minor revisions.
(1) In section 9 E-Waste Estimation and Prediction, authors are suggested to discussed the literature from accuracy of models, and moreover the relationship among various databases. These two aspects are of highly concerned in this research field.

(2) In the section of Recommendations, some general technologies are proposed to break through the bottlenecks of the application of e-waste management, more specific steps or valuable solutions are expected to read, especially comparing to other traditional strategies.

Author Response

Dear Sir/ Madam

The revised version of manuscript has been prepared and submitted for the further evaluation. The added content is added in red color., as in enclosed document.

Sincerely yours. 

Sudhanshu Joshi

Reviewer 2 Report

You have created a superb resource for this important area.   The location of data and approaches will be very helpful to others.  Hopefully! Your team can tackle the effectiveness of various models in the future and work to develop clear understanding by decision makers.

Author Response

(The authors gave the same response as above.)
